

# Internal structure of the action and acceptance questionnaire II (AAQ-II): evidence for a three-factor and bifactor model in two samples of university students

Daniel Núñez[1,2], César Villacura-Herrera[1], Javier Cubillos[3], Martín Donoso[3] and Álvaro I. Langer[2,4]

[1] Centro de Investigación en Ciencias Cognitivas, Faculty of Psychology, Universidad de Talca, Talca, Chile
[2] Millennium Nucleus to Improve the Mental Health of Adolescents and Youths (Imhay), Santiago, Chile
[3] Faculty of Psychology, Universidad de Talca, Talca, Chile
[4] Facultad de Psicología y Humanidades, Universidad San Sebastián, Santiago, Chile

Corresponding author
Álvaro I. Langer, alvaro.langer@gmail.com

## ABSTRACT

**Background:** Experiential avoidance (EA), defined as an inflexible behavioral pattern by which a person tries to avoid contact with private unpleasant situations, is considered a transdiagnostic variable associated with various psychopathological disorders. The Acceptance and Action Questionnaire II (AAQ-II) is a broadly used measure of EA. However, inconsistencies in the methods employed for examining its internal structure and the need for culturally validated measures in diverse populations highlight the need for further study. We conducted two studies in Spanish-speaking university students (nStudy1 = 829; nStudy2 = 830) to determine the validity and reliability of the AAQ-II.

**Methods:** In the first study, we examined the questionnaire through exploratory factor analysis (EFA). In the second study, we conducted confirmatory factor analysis (CFA) to examine how well these factor structures fit our data.

**Results:** We suggested a three-factor structure (namely: painful memories, emotional distress, and affective control), in contrast with the unidimensional model consistently reported in the literature. The three-factor model showed a better fit overall. Additionally, we tested a bifactor model to determine the feasibility of a general factor. Our results suggested that this general factor explained a high amount of the variance within the model when compared to the three independent factors. In terms of convergent validity, significant positive associations were found between EA and suicidal ideation, depressive and anxious symptoms, while negative associations were found with psychological well-being. The factor structures tested showed good internal consistency indices ($\omega = 0.853–0.918$; $\alpha = 0.853–0.918$).

**Discussion:** We discuss our results in terms of their methodological and practical implications for the use of the AAQ-II.

## INTRODUCTION

Experiential avoidance (EA) refers to the recurring attempts to suppress unpleasant thoughts, emotions, memories, or bodily sensations, even when this leads to actions that are inconsistent with personal values and goals (*Hayes et al., 1996*, *2004*). Although this regulatory strategy provides short-term relief (*Nielsen, Sayal & Townsend, 2016*), it is regarded as a transdiagnostic process associated with increased psychological distress, anxiety, depression, substance abuse (*Mellick et al., 2019*; *Ruiz et al., 2013*; *Spinhoven et al., 2016*), suicide ideation (*Angelakis & Gooding, 2021*; *Chou, Yen & Liu, 2018*), lower quality of life (*Hayes et al., 2004*), lower levels of wellbeing (*Machell, Goodman & Kashdan, 2014*), and interpersonal problems in clinical (*Akbari et al., 2022*) and general populations (*Brereton & McGlinchey, 2019*; *Martínez-Rubio et al., 2023*). This illustrates the importance of having access to measurement tools that can reliably and validly assess EA for research and clinical purposes, which is particularly relevant for Acceptance and Commitment Therapy (ACT; *Hayes, Strosahl & Wilson, 1999*).

The Acceptance and Action Questionnaire (AAQ; *Hayes et al., 2004*) has been one of the more often used tools for assessing EA, being adapted for multiple contexts and populations. Its psychometric properties, however, have been widely criticized (*Rochefort, Baldwin & Chmielewski, 2017*), showing inconsistencies concerning its factor structure and reliability indices when applied in different samples or contexts (*Bond et al., 2011*). To address the shortcomings of the original questionnaire, a seven-item revised version was developed by *Bond et al. (2011)*, known as the Acceptance and Action Questionnaire II (AAQ-II).

In recent years, the AAQ-II has been regarded as the main instrument for measuring EA (*Rochefort, Baldwin & Chmielewski, 2017*; *Mellin & Padrós, 2021*), showing good psychometric properties and good stability across different groups (*Bond et al., 2011*; *Ong et al., 2019a*). This has contributed to its use in a wide variety of countries and settings; being translated and adapted for use in Spanish (*Ruiz et al., 2013*), Portuguese (*Pinto-Gouveia et al., 2012*), Greek (*Karekla & Michaelides, 2017*), and Chinese (*Zhang et al., 2014*) populations, among others. However, the evidence on both its construct and discriminant validity is not clear (*Kashdan et al., 2020*). *Rochefort, Baldwin & Chmielewski (2017)* found that the AAQ-II functions as a measure of neuroticism/negative affect, while *Tyndall et al. (2019)* reported that the AAQ-II seemed to assess a construct somewhere between experiential avoidance and distress, which limited its ability to properly discriminate. This highlights the current need for further exploration of the AAQ-II as a valid and accurate measure for assessing the underlying processes of ACT (*McLoughlin & Roche, 2022*).

In terms of its internal structure, one of the main attributes of the AAQ-II is its unidimensionality. The internal structure of the test suggests the existence of a single underlying factor, which has been consistently reported in several studies (*Mellin & Padrós, 2021*). However, these studies possess some methodological issues that need to be addressed. First, studies exploring the underlying factor structure of the measure have usually performed principal components analysis (PCA) instead of exploratory factor

analysis (EFA). Although researchers commonly tend to use these analyses for the same purpose, both are conceptually distinct methods and thus have different aims. While the objective of PCA is to reduce data to the smallest possible set of components, EFA aims to identify underlying latent factors within the data (*Alavi et al., 2020*). Using PCA and EFA indistinctly or assuming that PCA is a method within EFA are common misconceptions that methodologists have addressed many times in the literature (*Bandalos & Boehm-Kaufman, 2009*; *Costello & Osborne, 2005*; *Fabrigar et al., 1999*; *Schreiber, 2021*).

Second, the recurring use of the Kaiser criterion as the method for determining the number of factors to be retained is currently viewed as an outdated strategy (*Howard, 2015*; *Rogers, 2022*). Under this method (*Kaiser, 1960*), all potential factors are examined and only those with eigenvalues equal or greater to 1 are preserved (this meaning that the retained factors should account for the equivalent variance of at least 1 item each). While this strategy is widely used for its simplicity, it is no longer recommended (*Fabrigar et al., 1999*; *Howard, 2015*; *Rogers, 2022*). Also, it has proven to have several weaknesses, such as over and underestimation of factors in presence of large or reduced sample sizes, number of items or factor loadings (*Fabrigar et al., 1999*; *Horn, 1965*; *Ruscio & Roche, 2012*; *Zwick & Velicer, 1986*). Due to this, alternative methods have been proposed for determining the number of factors in EFA.

One of the most robust and recommended methods is parallel analysis, which compares the observed eigenvalues with eigenvalues from multiple randomly generated datasets of the same size, aiming to retain only meaningful factors that explain more variance that would be expected from chance alone (*Horn, 1965*; *Silverstein, 1987*). This addresses the two main problems with the Kaiser criterion: Parallel analysis is less sensitive to sample size and number of items, as its comparison with random data reduces the likelihood of retaining factors based on mere chance or sampling errors, thus helping to identify factors which reflect more structure that would be expected under random conditions (*Hayton, Allen & Scarpello, 2004*; *Zwick & Velicer, 1986*).

Nevertheless, factor retention methods such as Kaiser criterion are still being widely used by researchers. This has also been the case for validation studies of the AAQ-II, where this technique has been consistently employed. Table 1 presents a brief review summarizing previous studies examining the internal structure of the AAQ-II and their methods. We conducted two studies. The first was aimed at exploring the underlying factor structure of the measure, also seeking to address the methodological inconsistencies existing in the literature by employing more robust and updated statistical techniques along with replicating findings from prior studies. The second study was aimed to confirm the validity, reliability, and overall psychometric properties of the AAQ-II. Both studies were conducted in two different samples of Spanish-speaking university students. We discuss our results in terms of their methodological and practical implications for the use of the AAQ-II as an assessment tool in future research and clinical practice.

**Table 1 Summary of previous validation studies.**

| Study | Sample | Extraction | Rotation | Factors | CFA | Initial fit | Notes |
|---|---|---|---|---|---|---|---|
| Bond et al. (2011) | USA university students, patients and bank employees | EFA+PA | Oblimin | Two, then one | Yes | Good | The authors initially found a two-factor structure on a 10-item version, but the second factor explained too little variance and was committed. Thus, after removal of some items with low factor loadings, a seven-item version was considered stronger. However, EFA was not tested based on these seven items only. A unidimensional structure was assumed. |
| Gloster et al. (2011) | Clinical-nonclinical German population. | — | — | One | Yes | Poor | The authors assume a single-factor structure using the seven-item version. CFA shows poor fit but improved by modification indices examination. MI was assessed and fully confirmed between subsamples of the study. |
| Pinto-Gouveia et al. (2012) | Portuguese general population | (i) PCA +KC; (ii) PCA +PA | Oblimin | (i) Two; (ii) One | Yes | Good | The authors initially found a two-factor structure using the seven-item version, but similar to the original study, they decided to move on to a single-factor structure due to the second factor explaining little variance. Is it worth noting that the authors state that they replicate the same methods as Bond et al. (2011), however they perform PCA instead of EFA. |
| Ruiz et al. (2013) | Patients and university students | PCA+KC | n/r | One | No | — | The authors found a single-factor structure using the seven-item version. No CFA was computed. |
| Pennato et al. (2013) | Italian adults | (i) EFA +PA; (ii) VSS; (iii) MAP | Oblimin | (i) Three; (ii) One; (iii) One | No | — | The authors report a KMO of 0.79, which is lower than expected for sample adequacy. They also report that item 5 "emotions cause problems in my life" showed low loadings for all factors. The single-factor solution showed higher loadings. Three-factor solution explained 64% of the variance, while the single-factor solution explained 42% of the variance. |
| Costa et al. (2014) | Clinical-nonclinical Portuguese population | — | — | One | Yes | Good | The authors assume a single-factor structure using the seven-item version. MI was assessed but was not confirmed for clinical-nonclinical samples. |
| Meunier et al. (2014) | Turkish university students | — | — | One | Yes | Poor | The authors assume a single-factor structure using the seven-item version. CFA shows poor fit but improved by modification indices examination. |
| Ghasemi et al. (2014) | Iranian university students | EFA+KC | Varimax | One | Yes | Acceptable | The authors found a single-factor structure using the seven-item version. CFA shows overall good fit except in RMSEA. Fit was improved by modification indices examination. |
| Zhang et al. (2014) | Chinese college students | EFA+PA/ KC | n/r | One | Yes | Poor | The authors found a single-factor structure using the seven-item version. While they report using PA, they also state that a single factor structure was found based on eigenvalues greater than 1, which correspond to KC. CFA shows poor fit but improved by modification indices examination. MI was assessed and fully confirmed for males/females. |

| Study | Sample | Extraction | Rotation | Factors | CFA | Initial fit | Notes |
|---|---|---|---|---|---|---|---|
| Wang et al. (2015) | Chinese university students | — | — | One | Yes | Acceptable | The authors assumed a single-factor structure. CFA results show good fit based on NI, IFI and CFI, and acceptable fit based on $\chi^2$/df and RMSEA. Article translated from Chinese. |
| Ruiz et al. (2016) | Colombian university students and clinical-nonclinical population | — | — | One | Yes | Acceptable | The authors assume a single-factor structure using the seven-item version. The authors explicitly state the use of polychoric correlations. CFA shows overall good fit except in RMSEA. Fit was improved by modification indices examination. |
| Yavuz et al. (2016) | Turkish patients and healthy controls | PCA+KC | — | One | Yes | Good | The authors found a single-factor structure using the seven-item version. CFA shows good fit. However, the authors perform both PCA and CFA on the same sample. |
| Karekla & Michaelides (2017) | Greek general population | — | — | One | Yes | Poor | The authors assume a single-factor structure using the seven-item version. CFA shows poor fit but improved by modification indices examination. MI was assessed and fully confirmed for males/females, but only confirmed at a metric level for clinical-nonclinical groups. |
| Eisenbeck & Szabó-Bartha (2018) | Hungarian general population | — | — | One | Yes | Poor | The authors assume a single-factor structure using the seven-item version. CFA shows poor fit but improved by modification indices examination. MI was assessed but only confirmed at a partial scalar level for males/females and partial metric level for treatment history/no treatment history groups. |
| Shari et al. (2019) | Malayan cancer patients | EFA+KC | n/r | One | No | — | The authors found a single-factor structure using the seven-item version. |
| Žuljević, Rakočević & Krnetić (2020) | Serbian adults | — | — | One | Yes | Poor | The authors assume a single-factor structure using the seven-item version. |
| Menéndez-Aller et al. (2021) | Spanish general population | — | — | One | Yes | Good | The authors assume a single-factor structure using the seven-item version. CFA shows good fit. |
| Paladines-Costa et al. (2021) | Ecuadorian university students | (i) PCA+KC; (ii) PCA+PA | Oblimin | (i) One; (ii) One | Yes | Poor | The authors found a single-factor structure using the seven-item version. This result was found through both PA and KC. CFA shows poor fit but improved by modification indices examination. MI was assessed and fully confirmed for males/females. |
| Mellin & Padrós (2021) | Mexican general population | PCA+KC | Varimax | Two, then one | Yes | Poor | Similar to Bond et al. (2011), the authors initially found a two-factor structure on a 10-item version but considered that the second factor explained too little variance and was difficult to interpret with only two items. CFA was performed on the seven-item version, showing poor fit. |

(Continued)

| Study | Sample | Extraction | Rotation | Factors | CFA | Initial fit | Notes |
|---|---|---|---|---|---|---|---|
| *Sadauska & Koļesovs (2021)* | Latvian adults | PCA+KC | Varimax | One | Yes | Acceptable | The authors found a single-factor structure using the seven-item version. CFA shows acceptable fit but improved by modification indices examination. |
| *Mohd Bahar et al. (2022)* | Malayan adults | EFA+n/r | n/r | One | No | — | The authors examined the underlying structure using EFA and found a single-factor structure. Methods for the number of factors extracted and rotation were not reported. |
| *Spencer et al. (2022)* | Hawaiian undergraduate students | EFA+n/r | n/r | (i) One; (ii) Three | Yes | (i) Poor, (ii) Poor | The authors first assume a single-factor structure. CFA shows poor fit overall. Then, they performed an EFA on Sample 1 and found a three-factor structure. CFA for this structure is tested on Sample 2 but shows poor fit. Then, the authors go back to the single-factor structure and examine modification indices, improving its fit. As a note, on their three-factor structure, the first factor consisted of a single item. Extraction and rotation methods are not detailed. |
| *Arias, Barraca & García (2023)* | Ecuadorian adults | EFA+PA | n/r | (i) One; (ii) Three | Yes | (i) Acceptable; (ii) Good | The authors first assume a unidimensional structure. CFA shows the seven-item version has acceptable fit indices. Then, the authors perform EFA and find a three-factor structure. This structure showed good and better fit indices all-round. However, it is important to note that the authors performed CFA on the same sample. |

**Note:**
EFA, Exploratory factor analysis; PCA, principal components analysis; PA, parallel analysis; KC, Kaiser criterion; VSS, very simple structure; MAP, minimum average partial test; CFA, confirmatory factor analysis; MI, measurement invariance.

# MATERIALS AND METHODS STUDY 1

## Participants and procedure

Students from two Chilean universities were recruited through an invitation on their institutional mailing list. Once written and informed consent was obtained, the participants completed a survey on mental health symptoms through the Qualtrics platform. Data was collected between June and November 2021. Participants did not receive any financial or academic incentive. According to a suicide risk protocol that was previously developed, all students who presented suicidal ideation in the last month and who expressed a possible suicidal intention in the following 12 months were contacted by telephone for a brief assessment of suicidality based on the Columbia-Suicide Severity Rating Scale (CSSR-S; *Posner et al., 2011*). Students showing imminent risk were referred to emergency services, and those who were at moderate or mild risk were referred to university support services.

Considering a conservative estimation suggested for factor analysis and using item observation ratios above 1:5, a sample size of >1,000 was targeted (*Costello & Osborne,*

2005; *MacCallum et al., 1999*; *Reio & Shuck, 2014*). The responses of 1,687 participants (female = 66.68%) aged between 18 to 36 years (M = 19.290; DE = 2.447) were collected. Based on the suggested age range of emerging adulthood (*Armett, 2015*), 28 participants (1.66% of the total sample) aged 30 or more were excluded from the study. Final sample size was 1,659 (M = 19.057; DE = 1.656; female = 66.79%). For cross-validation purposes, a randomly extracted subsample of 829 university students (49.97% of the final sample; female = 65.98%) aged between 18 to 29 years (M = 19.098; DE = 1.674) was used in Study 1.

Studies 1 and 2 were approved by the Scientific Ethics Committee of the National Health Service in Valdivia (N°075) and of the Universidad de Talca (03-2021). The study procedures were carried out in accordance with the Helsinki Declaration. All participants signed an informed consent for their participation in the studies.

## Measures

Participants responded to a web survey based on the World Health Organization's World Mental Health Undergraduate International Mental Health Initiative (WMH-ICS; *Cuijpers et al., 2019*). The authors have permission to use this instrument from the copyright holders. In addition, the Acceptance and Action Questionnaire-II (AAQ-II; *Ruiz et al., 2013*) was applied, which was not previously contained in the WMH-ICS web survey.

### Acceptance and action questionnaire II

The Acceptance and action questionnaire II (AAQ-II) is a seven-item self-report measure designed for the assessment of experiential avoidance and psychological inflexibility. Response options range from 1 ("never true") to 7 ("always true"). High scores on the AAQ-II are expected to be associated with an unwillingness to connect with unwanted emotions and thoughts, as well as an inability to be in the present and act according to personal values. The version used in the current study has been translated into Spanish and validated in the Spanish-speaking population, showing good internal consistency indices (between $\alpha = 0.75$ and $\alpha = 0.93$; *Ruiz & Luciano, 2009*; *Ruiz et al., 2013*). Item descriptions and its Spanish translations are available in Table S1.

## Data analysis

First, we tested the suitability of the sample for factor analysis through the Kaiser-Meyer-Olkin (KMO) and Bartlett's sphericity tests. Values over 0.80 for the KMO index and statistically significant results for the Bartlett's test are indicative of the suitability of the sample for factor structure examination (*Bartlett, 1954*; *Watkins, 2018*).

Second, we performed exploratory factor analysis (EFA) to examine the underlying structure of the AAQ-II in our sample. The number of factors to be extracted was determined through parallel analysis using a principal axis factoring estimator with an Oblimin rotation. Last, we examined the internal consistency of the AAQ-II in our sample through unidimensional reliability analysis, calculating McDonalds' omega ($\omega$) and Cronbach's alpha ($\alpha$) indices. Values over 0.70 would indicate a good internal consistency.

Complementarily, aiming to compare our results with the ones obtained in previous studies, we replicated their methods. First, we performed a principal components analysis

(PCA) where the number of factors to be retained was based on the Kaiser criterion (eigenvalues ≥ 1). Then, a PCA where the number of factors to be retained was based on parallel analysis. Next, an EFA where the number of factors to be retained was based on the Kaiser criterion. For both EFA and PCA, we based the analysis on the polychoric correlation matrix, as the response options for the measure are ordinal in nature and this method provides more accurate results in such cases (*Flora & Curran, 2004*; *Holgado-Tello et al., 2010*). Finally, although some studies based their analysis on a Varimax rotation, all our analyses used an Oblimin rotation as the measure is oblique in nature as defined by the authors (*Bond et al., 2011*). All analyses were performed using JASP v0.17.2 (*JASP Team, 2023*).

# RESULTS STUDY 1

## Sample adequacy tests

Results for the sample adequacy test were satisfactory, with both KMO (0.881) and Bartlett's sphericity test ($\chi^2_{Bartlett}$ = 4,890.967; $p < 0.001$) showing that this sample is suitable for examination through factor analysis.

## Exploratory factor analysis

To examine the underlying latent structure of the items, we conducted an EFA. Using a principal axis estimator and parallel analysis as the method for retaining factors, a three-factor structure emerged. The rotated solution of this structure explained 78.90% of the variance, with Factor 1 ("painful memories", comprised of items 1 and 4), Factor 2 ("emotional distress", comprised of item 5, 6 and 7) and Factor 3 ("affective control", comprised of items 2 and 3), explaining 27.10%, 26.90% and 24.90% respectively.

## Methods replication

Through following the statistical procedures of previous studies, we were able to replicate their results. By conducting either PCA or EFA based on Kaiser's criteria, a single-factor structure emerged. The same results were obtained when performing PCA based on parallel analysis. These rotated solutions explained 71.20% of the variance in the case of both PCA, and 66.50% for the EFA.

Internal consistency was found to be good for both the single- and three-factor structures, as indicated by McDonalds' ω and Cronbach α indices (>0.70). Results for all analyses are detailed in Table 2.

# MATERIALS AND METHODS STUDY 2

## Participants and procedure

For Study 2, the remaining random subsample of 830 university students (50.03% of the final sample; female = 67.59%) from two Chilean universities aged between 18 to 29 years (M = 19.017; DE = 1.637) was used. Participants did not receive any financial or academic incentive for participating in the study. For cross-validation purposes, students who had taken part in the previous study were deemed not eligible for participation.

**Table 2 Exploratory factor analysis, principal components analysis, sample adequacy tests and unidimensional reliability analysis.**

| | | Method | | | | | |
|---|---|---|---|---|---|---|---|
| | | EFA$_{PA}$ | | | PCA$_{KC}$ | PCA$_{PA}$ | EFA$_{KC}$ |
| | Items | F1 | F2 | F3 | C1 | C1 | F1 |
| Factor loadings | Item 1 | 0.925 | | | 0.819 | 0.819 | 0.782 |
| | Item 2 | | | 0.853 | 0.854 | 0.854 | 0.828 |
| | Item 3 | | | 0.909 | 0.853 | 0.853 | 0.827 |
| | Item 4 | 0.933 | | | 0.854 | 0.854 | 0.828 |
| | Item 5 | | 0.548 | | 0.891 | 0.891 | 0.881 |
| | Item 6 | | 0.674 | | 0.774 | 0.774 | 0.725 |
| | Item 7 | | 0.967 | | 0.855 | 0.855 | 0.829 |
| Unidimensional reliability analysis | ω | 0.904 | 0.857 | 0.853 | 0.918 | 0.918 | 0.918 |
| | α | 0.904 | 0.856 | 0.853 | 0.918 | 0.918 | 0.918 |

Note:
EFA, Exploratory factor analysis; PCA, principal component analysis; PA, parallel analysis; KC, Kaiser criterion; F, factor; C, component; ω, McDonalds' omega; α, Cronbach's alpha.

## Measures

### Experiential avoidance

Experiential avoidance (EA) was assessed through the Acceptance and Action Questionnaire II (AAQ-II: *Bond et al., 2011*) as described in Study 1.

Data concerning suicidal ideation and depressive/anxiety symptoms were collected as previously described in *Langer et al. (2024a)*, as follows:

### Suicidal ideation

Suicidal ideation (SI) was examined through two questions. The first one asks about recent SI (RSI): "*In the last 30 days, how often have you wished you were dead, wanted to go to sleep and never wake up, or had thoughts of suicide?*". Participants responded on a Likert-type scale ranging from 0 ("*never*") to 4 ("*always or almost always*"). The second question focused on future SI (FSI): "*In the next 12 months, how likely are you to attempt suicide?*". Participants responded on a Likert-type scale ranging from 1 ("*not at all likely*") to 4 ("*very likely*").

### Depressive symptoms

We assessed depressive symptoms within the past 30-days with the World Health Organization Composite International Diagnostic Interview (WMH-CIDI: *Kessler & Üstün, 2004*), which has been used as a screening tool for the mental health of students in multiple countries (*e.g.*, *Auerbach et al., 2018*). It uses a five-point scale, with 0 being "none of the time" and five being "always or most of the time". We used the following items: (i) you felt sad or depressed; (ii) you felt discouraged about how things were going in your life; (iii) you felt little or no interest or pleasure in things; and (iv) you felt bad about yourself or that you were not good enough or worthless. This scale showed good internal consistency in our sample (ω = 0.897; α = 0.894).

### Anxiety symptoms

We also employed the WMH-CIDI (*Kessler & Üstün, 2004*) to assess anxiety symptoms within the past 30-days by using the following items: (i) you felt anxious or nervous; (ii) you felt worried about several different things such as work, family, health or money; (iii) you felt more anxious, nervous or worried than other people in the same situation; (iv) you worried too much. Internal consistency indices were found to be good in our sample ($\omega = 0.906$; $\alpha = 0.905$).

### Psychological well-being

We used the short version of the Warwick-Edinburgh Mental Wellbeing Scales (SWEMWBS; *Stewart-Brown et al., 2009*; *Tennant et al., 2007*), a 7-item questionnaire. Response options range from 1 (*"none of the time"*) to 5 (*"all the time"*). Items cover different aspects of eudaimonic and hedonic well-being and are worded positively such as *"I've been feeling relaxed"*, and *"I've been dealing with problems well"*. The overall score is calculated by summing the scores for each item, with the minimum overall score being 7 and maximum score being 35. A higher score indicates a higher level of well-being (WB). Previous research found a unidimensional factor structure, along with strong internal consistency ($\alpha > 0.88$), construct validity and test–retest reliability ($ICC = 0.70$) in student samples of men and women and the general population (*Shah et al., 2021*; *Stewart-Brown et al., 2009*; *Sun et al., 2019*). Internal consistency for the WB scale was found to be good in our sample ($\omega = 0.874$; $\alpha = 0.872$).

## Data analysis

Based on the factor structures found in Study 1, we aimed to determine the best fitting factor structure of the AAQ-II. After ensuring the current sample adequacy for factor analysis through the KMO index and Bartlett's sphericity test, both the single- and three-factor structures were examined through confirmatory factor analysis (CFA). Both models were compared in terms of both absolute and relative fit indices, along with model parsimony. Furthermore, the feasibility of a general factor within the three-factor structure observed was tested through bifactor analysis.

Absolute fit indices include the chi-square divided by degrees of freedom ratio ($\chi^2/df$), the root mean square error of approximation (RMSEA) and the standardized root mean square residual (SRMR). Relative fit indices include the comparative fit index (CFI), Tucker-Lewis index (TLI), and goodness of fit index (GFI). Criteria for examining these indices is given for established cut-off values ($\chi^2/df < 5$; RMSEA $< 0.08$; SRMR $< 0.08$; CFI $\geq 0.90$; TLI $\geq 0.90$; GFI $\geq 0.95$; *Hu & Bentler, 1999*). Model parsimony was examined through the Akaike information criterion (AIC) and the Bayesian information criterion (BIC), where lower values in both indices provide evidence of better overall fit and parsimony.

It has been proposed that traditional fit indices may tend to be biased in the context of bifactor analysis due to the increased number of parameters, increasing the likelihood of these models to be overfitted (*Eid et al., 2017*; *Greene et al., 2019*), although this has been a topic of discussion in recent years (*Bader & Moshagen, 2022*). Aiming to address this, we
calculated specific bifactor statistical indices to provide a more robust estimate of the overall fit and performance of this model. The Omega total (ωt) is an overall estimate of the model reliability, indicating the proportion of variance in the scores that can be attributed to a combination of the general and independent factors. The Omega hierarchical index (ωh) provides a proportion of variance attributable specifically to the GF. Values below 0.50 may suggest that the variance within the model is explained mainly by the independent factors alone (*Revelle & Zinbarg, 2008*; *Reise, Bonifay & Haviland, 2013*). Complementarily, we calculated the explained common variance (ECV), an estimate of the proportion of common variance explained by the GF in relation to the independent factors. We also obtained the specific ECV for each factor (S-ECV), to estimate their overall uniqueness within the bifactor model (*Bentler, 2008*; *Reise et al., 2012*). The percentage of uncontaminated correlations (PUC) reflects the proportion of inter-item correlations that are explained solely by the GF, with values above 0.80 indicating a predominant influence of this factor (*Bonifay et al., 2015*; *Reise et al., 2012*). Finally, the factor determinacy (FD) and construct replicability (H) indices were computed. The FD index estimates how well the factors within the model are determined by the indicators (items), with values closer to 1 indicating better determinacy (*Rodriguez, Reise & Haviland, 2016*). The H index provides an estimate of the internal robustness of a factor or dimension, with values closer to 1 indicating a structure that is more likely to be replicated across samples (*Hancock, 2001*; *Hancock & Mueller, 2001*).

Next, we assessed convergent validity of the AAQ-II by examining its associations with RSI, FSI, DP and AN scales of the WMH-CIDI and with the WB scale. Associations were examined through correlation analysis using the three factors and the total score of the AAQ-II. Internal consistency of the AAQ-II was also examined through unidimensional reliability analyses, calculating McDonalds' omega (ω) and Cronbach's alpha (α) indices. Finally, all analyses were performed using JASP v0.17.1.0 (*Love et al., 2019*), and R with the *lavaan* (*Rosseel, 2012*), *performance* (*Lüdecke et al., 2021*), and *BifactorIndicesCalculator* (*Dueber, 2019*) packages.

## RESULTS STUDY 2

### Sample adequacy tests!

Results for the sample adequacy tests were satisfactory, with both KMO and Bartlett's sphericity test showing that this sample is suitable for examination through factor analysis (KMO = 0.882; $\chi^2_{Bartlett}$ = 5,004.186; $p < 0.001$).

### Confirmatory factor analysis

Our findings indicate that, when comparing both, the three-factor structure demonstrated a better fit all-round when directly compared to the single-factor structure in terms of absolute, relative and parsimony indices. However, neither of the structures showed an ideal fit in terms of the $\chi^2$/df ratio or RMSEA. The bifactor model (Fig. 1) showed excellent fit indices overall when compared with the three- and single-factor structures. It also showed better model parsimony overall and good $\chi^2$/df ratio and RMSEA. Results for all the models tested are detailed in Table 3.

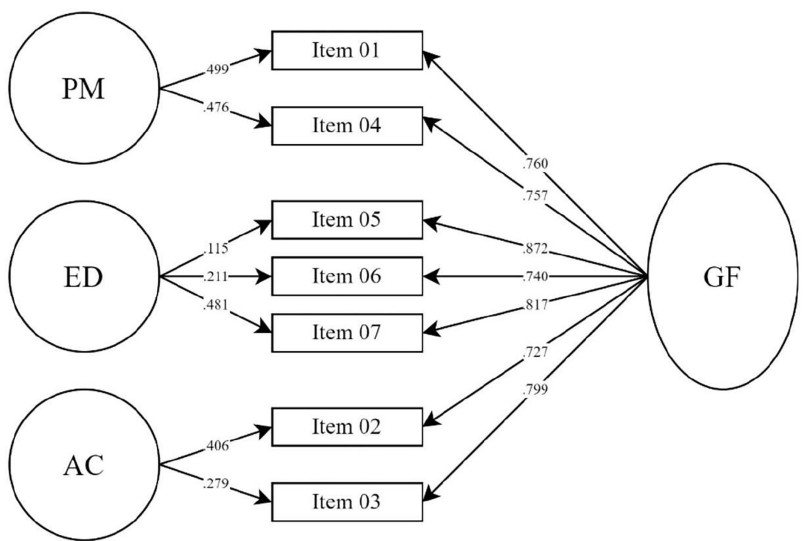

**Figure 1 Path diagram of the bifactor AAQ-II model.** PM, Painful memories; ED, emotional distress; AC, affective control; GF, general factor.

**Table 3 Confirmatory factor analysis and bifactor analysis results.**

|  | Absolute fit | | | | | Relative fit | | | Parsimony | |
| --- | --- | --- | --- | --- | --- | --- | --- | --- | --- | --- |
|  | $\chi^2$ | df | $\chi^2$/df | RMSEA | SRMR | CFI | TLI | GFI | AIC | BIC |
| Three-factor | 97.350 | 11 | 8.850 | 0.097 | 0.019 | 0.980 | 0.961 | 0.987 | 19,390.632 | 19,503.947 |
| Single-factor | 472.335 | 14 | 33.738 | 0.199 | 0.046 | 0.892 | 0.839 | 0.944 | 19,760.069 | 19,792.531 |
| Bifactor | 77.898 | 21 | 3.709 | 0.110 | 0.018 | 0.983 | 0.950 | 0.975 | 19,372.776 | 19,471.926 |

**Note:**
RMSEA, Root mean square error of approximation; SRMR, standardized root mean square; CFI, comparative fit index; TLI, Tucker-Lewis index; GFI, goodness of fit index; AIC, Akaike information criteria; BIC, Bayesian information criteria.

As previously stated, traditional fit indices may tend to overestimate the fit of bifactor models due to the increased number of estimated parameters, thus providing a potentially biased representation of the model's true performance. Calculation of Omega total and ECV indices for the general factor (GF) supported this, with high values suggesting a dominance of this factor within the proposed model ($\omega_{Total} = 0.950$; $ECV_{GF} = 0.810$). Calculation of specific bifactor indices also follow this line, with the Omega hierarchical coefficient suggesting that the GF captures most of the variance ($\omega h_{GF} = 0.888$), and S-ECV values indicating that the independent factors explain a reduced amount of variance overall ($S\text{-}ECV_{PM} = 0.292$; $S\text{-}ECV_{ED} = 0.128$; $S\text{-}ECV_{AC} = 0.172$). In terms of determinacy and construct replicability, the three independent factors show moderate values, reflecting that the items are properly associated with the underlying constructs measured, and these three independent factors may not be as consistent or stable as expected ($FD_{PM} = 0.770$; $H_{PM} = 0.384$; $FD_{ED} = 0.723$; $H_{ED} = 0.265$; $FD_{AC} = 0.594$; $H_{AC} = 0.219$).

## Convergent validity

For convergent validity, five measures were considered: RSI, FSI, DP, AN and WB. Our results showed statistically significant and positive correlations between painful memories (PM) and RSI ($r_S = 0.289$), FSI ($r_S = 0.326$), DP ($r_S = 0.553$), and AN ($r_S = 0.534$). Similarly, emotional distress (ED) demonstrated positive correlations with RSI ($r_S = 0.399$), FSI ($r_S = 0.363$), DP ($r_S = 0.680$), and AN ($r_S = 0.608$). Affective control (AC) also showed positive correlations with RSI ($r_S = 0.344$), FSI ($r_S = 0.307$), DP ($r_S = 0.580$), and AN ($r_S = 0.566$). Finally, our analysis revealed negative statistically significant correlations between PM and WB ($r_S = -0.463$), ED and WB ($r_S = -0.653$), and AC and WB ($r_S = -0.503$).

Additionally, when examining the AAQ-II total score as a unidimensional measure, results were similar, showing positive correlations with RSI ($r_S = 0.396$), FSI ($r_S = 0.383$), DP ($r_S = 0.683$), and AN ($r_S = 0.639$), and a significant negative correlation with WB ($r_S = -0.618$). The $p$ values for correlation analyses were <0.001 in all cases.

## Reliability estimate

Our results indicate good internal consistency indices for both the three-factor structure ($\omega_{PM} = 0.896$; $\alpha_{PM} = 0.896$; $\omega_{ED} = 0.881$; $\alpha_{ED} = 0.881$; $\omega_{AC} = 0.818$; $\alpha_{AC} = 0.818$) and the single-factor structure ($\omega = 0.925$; $\alpha = 0.924$).

## DISCUSSION

We conducted two studies to provide evidence about psychometric properties of the AAQ-II in two samples of university students in Chile. Two factor structures for the AAQ-II were examined through CFA: The three-factor structure based on our analysis and the single-factor structure based on replication of methods by previous studies. In addition, a bifactor model was tested including a general factor (experiential avoidance) within the three-factor model. We found that a three-factor structure yielded the best fit and structural validity, also showing good reliability and convergent validity. These findings contrast prior research reporting an unidimensional structure of this measure (*Bond et al., 2011*, *Pinto-Gouveia et al., 2012*; *Ruiz et al., 2013*) but are in line with recent studies that have examined the underlying factor structure through currently recommended statistical procedures (*Pennato et al., 2013*; *Spencer et al., 2022*; *Arias, Barraca & García, 2023*). Additionally, the bifactor model showed better fit indices overall when compared with the three- and single-factor structures.

Based on our results, the bifactor structure provided a better representation of the underlying latent construct measured by the AAQ-II. This proposed model explained a higher amount of variance and had a better fit all-round when compared to the commonly assumed unidimensional model. Three-factor structures have been also found by *Pennato et al. (2013)*, and *Spencer et al. (2022)*. The former reported a three-factor model accounted for 64% of the variance, compared with the 42% accounted for by the unidimensional solution. The latter briefly examined a three-factor structure in which one factor was composed by a single item. Similarly, *Arias, Barraca & García (2023)* observed a three-factor configuration that accounted for 69.3% of the variance, compared with the

25.5% accounted for in the unidimensional model. In addition, studies performing confirmatory analyses for the single-factor structure tended to report poor fit indices overall (See Table 1). While these structures tend to slightly differ, a factor composed by items 1 and 4 remains constant. It is also worth mentioning that when using currently unsuitable methods to estimate the number of underlying factors (*Gaskin & Happell, 2014*; *Watkins, 2018*), we were able to replicate the unidimensional configuration in our sample.

The three-factor structure found by *Pennato et al. (2013)* and *Arias, Barraca & García (2023)* showed almost the same item distribution observed in our sample. The only difference was item 5 ("emotions cause problems in my life"), which the authors found to belong to the second factor, along with items 2 ("I'm afraid of my feelings") and 3 ("I worry about not being able to control my worries and feelings"). Regarding this, *Pennato et al. (2013)* reported this particular item to show low loadings on all factors. Our analysis also revealed that it showed the lowest factor loading, thus it is possible that it possesses certain ambiguity and may require further examination. A complementary examination through CFA (see Table S2) allowed us to confirm two things: First, that the three-factor structure we propose in this study demonstrates better fit when compared to previous approaches. Second, that any of the three-factor structures shows better fit when compared to the unidimensional model. Our results from the bifactor analysis also demonstrated better fit overall, also suggesting the importance of understanding the AAQ-II not simply as a unidimensional solution, but a more complex structure with a general factor able to encompass the construct of experiential avoidance.

It should be noted that, while the unidimensional structure was initially reported and proposed by *Bond et al. (2011)*, who followed the same statistical procedures we used, their findings were mainly based on a 10-item version that showed to have two-factors. Their second factor, which was composed of three items, was removed due to explaining only 4.94% of the variance. However, the authors did not examine the underlying structure based on these seven items only. On the other hand, most subsequent studies did not follow the same procedures as the original authors, and either employed different data analysis strategies, or simply assumed a single factor structure without further examination.

Our findings add new insights for a more nuanced description of the construct and suggest that EA, as assessed by the AAQ-II, might not be unidimensional but a construct encompassing multiple processes, as previously stated (*Hekmati et al., 2021*). The content of each factor was quite similar to those reported by *Arias, Barraca & García (2023)* in adults from Ecuador. Although all factors might serve the same psychological function (*i.e.*, experiential avoidance), they seem to reflect slightly different facets of the latent trait. The first factor was linked to painful memories, which emphasize the potential negative impact of prior stressful events, experienced as life-affecting memories in the present. This could reflect the associations shown between EA and traumatic-based events (*Orcutt, Reffi & Ellis, 2020*). Interestingly, this factor contains item 4. When examined under the Item Response Theory—a psychometric approach that complements Classical Test Theory by assessing each item's capacity to discriminate levels of the latent trait (*Reise & Waller,*

*2008*), it has been proven to be the most significant item for clinical purposes, given that it needs a higher level of the latent trait to be endorsed by participants (*Langer et al., 2024b*; *Ong et al., 2019b*). This means that this item can work as an indicator within the construct, showing a greater discriminative ability to identify participants exhibiting past memories that impact its present state. The second factor (composed by items 5, 6 and 7) was related to the interference of emotions in the desired life when compared to oneself or others. This factor contains items that reflect a general evaluation of life (item 6: "It seems like most people are handling their lives better than I am") or general and unspecific consequences of emotions on it (item 5: "emotions cause problems in my life"). Moreover, item 6 has consistently shown the lowest discriminative capacity of the scale (*Menéndez-Aller et al., 2021*; *Ong et al., 2019b*), while item 5 has presented a good ability to discriminate EA (*Menéndez-Aller et al., 2021*).

The third factor, composed by items 2 and 3, refers to difficulties in self-regulation and explicitly addresses the fear of experiencing feelings and not being able to handle them, suggesting that coming into contact with some internal events can be an aversive experience that should be controlled. As stated by *Ong et al. (2019b)*, items reflecting broad language, such as feelings or emotions have shown less discriminative capability of the latent trait than those using specific terms like worries or memories. Thus, these items might not be directly assessing the specific nature of EA and might be capturing variance from other related constructs. These results must be interpreted in the context of the existing debate in the literature about whether the AAQ-II does assess EA or other related constructs such as neuroticism, psychological distress or negative affect (*Wolgast, 2014*; *Ruiz et al., 2013*; *Rochefort, Baldwin & Chmielewski, 2017*; *Tyndall et al., 2019*). While our results showed that under recommended statistical methods the measure shows a replicable and coherent structure, further research is still needed to fully clarify its construct validity across both clinical and research contexts.

Concerning the convergent validity of the AAQ-II, our results support recent research showing associations between EA, depressive and anxiety symptoms (*Akbari et al., 2022*) and suicide ideation in university students (*Chou, Yen & Liu, 2018*; *Hekmati et al., 2021*). In addition, the negative associations between EA and perceived well-being fit with prior research revealing associations between EA and lack of meaning in life (*Pavlacic, Schulenberg & Buchanan, 2021*), negative affect and less enjoyment of life events (*Machell, Goodman & Kashdan, 2014*), and poorer quality of life in college students (*Coutinho, Trindade & Ferreira, 2019*). Overall, this evidence suggests good convergent validity for both the total scale and each subscale. Nevertheless, considering the evidence suggesting that EA could be a context-specific regulatory strategy (*Machell, Goodman & Kashdan, 2014*), further examination on the interactions with other psychological constructs is still needed.

Our finding revealing a three-factor and bifactor configuration allows for a better understanding of different facets of EA through a very brief assessment. This is in line with literature suggesting a more comprehensive assessment of EA (*Spinhoven et al., 2016*). This has potential clinical and research implications. An accurate distinction of which

avoidance facet is prominent could be helpful to define specific therapeutic targets, objectives, and strategies. Moreover, this provides a parsimonious measure to assess associations among different facets of EA and psychopathological and wellbeing domains in different contexts and populations. In terms of its practical use, clinicians and counselors could continue to evaluate EA through a single score based on the general factor, keeping in mind that it encompasses three underlying dimensions rather than being the result of a unidimensional solution. We are aware that the AAQ-II has been used both as a measure of psychological inflexibility and experiential avoidance rather than experiential avoidance only. This is a current and relevant issue regarding its validity to accurately assess ACT psychological process (*Kashdan et al., 2020*; *McLoughlin & Roche, 2022*; *Ruiz et al., 2024*). Our findings provide additional insights to contribute to this scientific controversy.

Some limitations deserve mention. First, because of our cross-sectional design, we did not address the temporal stability of the scale. Second, we did not assess other symptoms, emotional regulatory processes and psychological distress potentially associated with EA. Third, we only included first year students and used a non-probabilistic sampling method. Therefore, our findings can be interpreted with caution in terms of their replicability to other populations. In addition, it should also be noted that as the present study was conducted in a Spanish-speaking population, culturally shaped response styles could be a relevant factor to consider. Research has shown that some culturally shaped response patterns linked to emotion expression can have an influence in factor loadings in self-report measures (*Harzing, 2006*; *van de Vijver & Leung, 2001*). Studies on the cultural invariance of the AAQ-II considering countries from different continents are still needed to address this.

Furthermore, it may rise as a concern that two factors of the proposed structure are composed by two items only, contrary to the usually recommended three-items per factor rule of thumb. However, researchers should be aware that this derives from a comment in *Tabachnick & Fidell (2013)*, who only suggested that two item factors may struggle to accurately represent the underlying construct it aims to measure. If two items are deemed sufficient for this purpose, then two items may suffice (*Worthington & Whittaker, 2006*). Thus, the theoretical interpretability and meaningfulness of the factor should always be the most important retention criteria, regardless of the number of items included in each factor.

## CONCLUSIONS

In conclusion, our results show a bifactor model for the AAQ-II, which contrasts with prior evidence of a unidimensional structure. According to the intended use of the scale, both a single score or split into its three dimensions are suitable for clinical and research purposes. We identified items that consistently have shown loading and interpretability issues. The scale seems to be composed of items with different levels of proximity to EA/psychological inflexibility, which needs to be taken into account when the construct is measured through this tool.

## ACKNOWLEDGEMENTS

The survey was carried out in conjunction with the WHO World Mental Health International College Student Initiative (WHO WMH-ICS). We thank the staff of the WMH Data Analysis Coordination Centre for assistance with instrumentation and data analysis. A complete list of all within-country and cross-national WMH-ICS publications can be found at http://www.hcp.med.harvard.edu/wmh/. We would like to thank the university students who participated in this study.

### Funding

The current study was funded by ANID—Millennium Science Initiative Program—NCS2021_081; Vicerrectoría de Investigación y Doctorados de la Universidad San Sebastián—Fondo USS-FIN-25-APCS-18; and the Programa de Investigación Asociativa (PIA) en Ciencias Cognitivas (RU-158-2019), Centro de Investigación en Ciencias Cognitivas (CICC), Universidad de Talca. Álvaro Langer and Daniel Nunez are supported by ANID-FONDECYT regular N° 1221034 and 1210093 respectively. The funders had no role in study design, data collection and analysis, decision to publish, or preparation of the manuscript.

### Grant Disclosures

The following grant information was disclosed by the authors:
ANID—Millennium Science Initiative Program: NCS2021_081.
Vicerrectoría de Investigación y Doctorados de la Universidad San Sebastián—Fondo USS-FIN-25-APCS-18.
Programa de Investigación Asociativa (PIA) en Ciencias Cognitivas: RU-158-2019.
Centro de Investigación en Ciencias Cognitivas (CICC).
Universidad de Talca.
Álvaro Langer and Daniel Nunez, ANID-FONDECYT: N° 1221034 and 1210093.

### Competing Interests

The authors declare that they have no competing interests.

### Author Contributions

- Daniel Núñez conceived and designed the experiments, authored or reviewed drafts of the article, and approved the final draft.
- César Villacura-Herrera performed the experiments, analyzed the data, prepared figures and/or tables, authored or reviewed drafts of the article, and approved the final draft.
- Javier Cubillos performed the experiments, authored or reviewed drafts of the article, and approved the final draft.
- Martín Donoso performed the experiments, authored or reviewed drafts of the article, and approved the final draft.
- Álvaro I. Langer conceived and designed the experiments, authored or reviewed drafts of the article, and approved the final draft.

## Human Ethics

The following information was supplied relating to ethical approvals (*i.e.*, approving body and any reference numbers):

The study was approved by the scientific ethical committee of the National Health Service in Valdivia (n° 075) and the Universidad de Talca (03-2021). The study procedures were carried out in accordance with the Helsinki Declaration.

## Data Availability

The raw measurements and data are available in the Supplemental Files.

## Supplemental Information

Supplemental information for this article can be found online at http://dx.doi.org/10.7717/peerj.19620#supplemental-information.

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
