# Peer review of "Internal structure of the action and acceptance questionnaire II (AAQ-II): evidence for a three-factor and bifactor model in two samples of university students"

_PeerJ, doi:10.7717/peerj.19620_

## Round 0.1 · original submission · Minor Revisions

Overall, the feedback is highly positive, with both reviewers commending the clarity of writing, robustness of the methodology, and the valuable contribution your study makes to the literature on the AAQ-II.

To move forward with the publication process, please address the following minor revisions:

Address minor language edits: Reviewer 2 has noted several small grammatical issues (e.g., corrections to phrasing and punctuation). Please revise accordingly throughout the manuscript.

Expand discussion on AAQ-II validity: Both reviewers suggested strengthening the limitations or discussion sections by more clearly acknowledging the ongoing debate about what the AAQ-II measures—specifically, its potential overlap with general psychological distress or psychological inflexibility. I note that Reviewer 2 has provided several references for you to consider.

Clarify the contribution: Reviewer 2 recommends a clearer statement of how your findings build upon or differ from previous studies, such as Ong et al. (2019).

Correct reporting conventions: Please revise any instances of reporting p-values as “p = .000” to “p < .001” to align with conventional statistical reporting.

Optional but recommended: Consider adding a brief note in the limitations about potential cultural influences on response patterns, as suggested by Reviewer 1.

Please submit a revised version along with a response letter outlining how you addressed each point.

Kind regards,
Shane Rogers

**Language Note:** The review process has identified that the English language must be improved. PeerJ can provide language editing services - please contact us at [email protected] for pricing (be sure to provide your manuscript number and title). Alternatively, you should make your own arrangements to improve the language quality and provide details in your response letter. – PeerJ Staff

**Staff Note:** Is there a typo in the title? Should "throughout" be "thorough"?

Reviewer 1 ·

Basic reporting

The manuscript is well written in professional, clear, and precise English.

The introduction effectively situates the research within the existing literature on experiential avoidance (EA) and the AAQ-II. The authors provide a comprehensive review of prior studies and highlight key gaps in the literature.

Tables and figures are clear, well-labeled, and informative.

Experimental design

The research question is well-defined, addressing a critical gap regarding the internal structure of the AAQ-II in Spanish.

The study design is robust, employing both exploratory factor analysis (EFA) and confirmatory factor analysis (CFA). The decision to use parallel analysis rather than the Kaiser criterion is commendable and aligns with current best practices.

The sample size is adequate for factor analysis, and the use of two independent samples strengthens the findings.

Validity of the findings

The statistical methods are appropriate and rigorously applied. And the use of a bifactor model is a valuable addition, as it helps reconcile the presence of multiple factors with the broader construct of EA.

The results are interpreted thoughtfully, with clear links back to the research question and theoretical framework. The discussion section carefully weighs the implications of a three-factor vs. unidimensional structure.

Convergent validity is well established, with appropriate correlations to related constructs (e.g., depression, suicidal ideation, well-being).

Additional comments

The manuscript contributes valuable insights into the psychometric properties of the AAQ-II, especially in non-English-speaking contexts. The use of contemporary statistical techniques enhances the study’s credibility.

The limitations section is honest and transparent, but the authors might consider adding a note on potential cultural influences on response styles, as this could affect factor loadings.

While the AAQ-II is widely used to assess experiential avoidance, its validity remains debated. Some studies suggest it may capture general psychological distress or negative affect more than experiential avoidance itself. Given this ongoing debate, the authors might consider discussing this broader context in their limitations or implications section, acknowledging that while their findings strengthen the AAQ-II’s structural validity, its ability to purely measure experiential avoidance remains an open question.

Reviewer 2 ·

Basic reporting

The paper is well-written and professional, with just a few minor typos.


Lines 137-139 - 'and conducted by the Helsinki Declaration' should be 'and conducted in accordance with the Helsinki Declaration'; 'According to a suicide protocol was previously developed' is missing 'that' (i.e., that was previously developed).

"The studies 1 and 2 were approved" would be better as 'Studies 1 and 2 were approved'

Line 279 "In our sample. internal consistency indices" - remove the full stop.

Line 339 - "Finally, All analyses" should be lower case, all analyses.

The literature review is fine for analytical choices but is rather limited in comparison to paper's such as Rochefort et al. (2018) and Tyndall et al. (2019), in terms of explaining to the reader the importance of the issue of AAQ's construct and discriminant validity. Indeed, those papers are cited but there is no real engagement with them and. Addressing this gap will help present an even more compelling rationale for the need for this present study to inform and advance the literature. There have been other critiques of the AAQ-II that could be referred to in Introduction such as that of McLoughlin, S., & Roche, B. T. (2023). ACT: A Process-Based Therapy in Search of a Process. Behavior therapy, 54(6), 939–955. https://doi.org/10.1016/j.beth.2022.07.010 and of the AAQ-II's capacity to measure psychological flexibility such as Kashdan, T. B., Disabato, D. J., Goodman, F. R., Doorley, J. D., & McKnight, P. E. (2020). Understanding psychological flexibility: A multimethod exploration of pursuing valued goals despite the presence of distress. Psychological assessment, 32(9), 829–850. https://doi.org/10.1037/pas0000834.

I think the authors could add in a section with relevant citations stating they are aware that the AAQ-II is often used/viewed as a measure of psychological inflexibility rather than experiential avoidance only, as this is a common critique by advocates of the AAQ-II who dismiss scientific critiques of the instrument, and strongly stating their case for their focus on experiential avoidance. I would also more clearly state what Ong et al.'s (2019) Item Response Theory paper found in more simple terms for your reader, and what you are offering here that is different to or above and beyond their work for implications in understanding the psychometric properties and validity of the AAQ-II for measuring what it purports to measure.

Experimental design

I am generally satisfied with the design and that analytical decisions and choices were clearly explained throughout. I had a few thoughts around whether a different approach could be taken at times but on reflection I will not detail them here as the authors are clear in their rationale for the stances they have taken an justified it in the literature. The research question is still meaningful despite the AAQ-II having been examined and critiqued in multiple peer-reviewed articles previously in the literature. The paper evidences the rigour in the investigation and I expect that the methods (the EFA, CFA) are replicable based on the detail provided.

Validity of the findings

The data appear robust and statistically sound and controlled with clear rationale provided for each analytic choice and power analyses conducted.

General point on convention - it's risky to state p = .000 as has been reported a few times in the manuscript and in such instances it might be safer to state p < .001.
"Results for the sample adequacy tests were satisfactory, with both KMO and Bartletts
sphericity test showing that this sample is suitable for examination through factor analysis (KMO=3.882;

Additional comments

This is an interesting study that I would like to see published once minor changes have been addressed as it challenges the generally believe that the AAQ-II is a unitary factor instrument.

---

## Round 0.2 · Minor Revisions

Thank you for submitting your revised manuscript to PeerJ. The reviewer has recommended publication, but I just have two small suggestions (title and abstract) that I think would assist readers.

1) I think the title can be simplified to "Internal structure of the Action
and Acceptance Questionnaire II (AAQ-II): Evidence for a
three-factor and bifactor model in two samples of university
students"

2) In the results subsection of the abstract, I think it would be helpful to name the three factors identified (so that readers are aware right from the start of the paper that the identified subscales tap into painful memories, emotional distress, and affective control)

Thank you again for submitting your work to PeerJ.

Reviewer 2 ·

Basic reporting

I am satisfied with the revision and all the supplementary files provided. Well done. The literature reviewed is more grounded in context.

Experimental design

Yes, a systematic and logical analytical approach was followed and reported.

Validity of the findings

I think the findings are important and would expect to see this 3-factor structure replicated in future studies.

Additional comments

I am sure the copy editor will pick up the remaining minor issues with referencing and APA style (e.g., using '&' outside of parentheses; not presenting citations in alphabetical order within parentheses; missing citations in the text that are not in the Reference List such as Doorley & McKnight, 2020; and minor APA errors in the Reference List itself)

---

## Round 0.3 · accepted · Accept

Thank you for addressing my very minor comments. I believe this paper is ready for publication. Thank you for submitting your work to PeerJ.